# Galactomannans for Entrapment of *Gliomastix murorum* Laccase and Their Use in Reactive Blue 2 Decolorization

Itzel C. Romero-Soto [1], Raúl B. Martínez-Pérez [2], Jorge A. Rodríguez [2], Rosa M. Camacho-Ruiz [2], Alejandra Barbachano-Torres [2], Martha Martín del Campo [1], Juan Napoles-Armenta [3], Jorge E. Pliego-Sandoval [4], María O. Concha-Guzmán [1],* and María Angeles Camacho-Ruiz [1],*

[1] Laboratorio de Investigación en Biotecnología, Centro Universitario del Norte, Universidad de Guadalajara, Colotlán 46200, Jalisco, Mexico; missromero89@hotmail.com (I.C.R.-S.); martha.martindelcampo@academicos.udg.mx (M.M.d.C.)

[2] Biotecnología Industrial, Centro de Investigación y Asistencia en Tecnología y Diseño del Estado de Jalisco (CIATEJ), Zapopan 45019, Jalisco, Mexico; rbmperez@gmail.com (R.B.M.-P.); jrodriguez@ciatej.mx (J.A.R.); rcamacho@ciatej.mx (R.M.C.-R.); ale_b_t@hotmail.com (A.B.-T.)

[3] Facultad de Agronomía, Universidad Autónoma de Nuevo León, Francisco I. Madero S/N, Ex Hacienda el Cañada, Gral. Escobedo 66050, Nuevo León, Mexico; jnapolesarmenta@gmail.com

[4] Laboratorio de Ciencias de la Naturaleza. Centro Universitario del Sur, Av. Enrique Arreola Silva 883, Col Centro, Ciudad Guzmán 49000, Jalisco, Mexico; jorge.pliego@cusur.udg.mx

* Correspondence: mocg@cunorte.udg.mx (M.O.C.-G.); angeles_camacho@cunorte.udg.mx (M.A.C.-R.)

**Abstract:** In the present study, a novel laccase from ascomycete *Gliomastix murorum* was produced in agro-industrial wastes and entrapped in galactomannan beads for Reactive Blue 2 (Rb-2) decolorization. The maximum laccase production in agave bagasse-based medium occurred at 72 h (1798.6 UL$^{-1}$). Entrapped laccase decolorized >80% of 0.5 mM Rb-2 in 2 h without the addition of redox mediator. $K_m$ for Rb-2 substrate was 1.42 mM, with a $V_{max}$ of 1.19 µmol min$^{-1}$ for entrapped laccase. Galactomannan matrices produce stability to acid pH (2–5) tion in the elimination of different dyes, and results showedand temperatures from 20–70 °C. Reusability assays showed that entrapped laccase could retain efficient Rb-2 decolorization of >80% six times. In general, galactomannan used for entrapment of laccase provides economic advantages in large-scale wastewater treatment due to its natural origin and efficient results.

**Keywords:** galactomannan beads; ascomycete; laccase; dye decolorization; cibacron blue 3GA; immobilization

## 1. Introduction

Organic synthetic dyes represent one of biggest contamination problems generated by textile, paper, plastics, pharmaceutical, food, and cosmetics industries [1–3]. Dyes are discharged into aquatic environments, damaging ecosystems due to their toxicity. Consequently, there is a reduction in photosynthetic activity by the obstruction of solar rays. In addition, dyes represent a risk to human health due to their potential mutagenic and carcinogenic effects [4]. Reactive anthraquinoic dyes have a stable aromatic structure, which makes their elimination difficult. These dyes represent around 20–30% of dyes used worldwide [5]. They have a low fixation in cotton, wool garments, and others during the dyeing stage, which means around 8–35% of dyes applied are discharged into residual effluents [6,7].

Reactive Blue 2 (Rb-2) is an anthraquinoic dye, its structure is a triazine with amino and sulfate groups, giving it high stability and resistance to degradation by biological systems and chemical treatments [8,9]. Due to this problem, several efforts have been made to remove Rb-2 from the environment. Traditionally, different physical treatments have been proposed to eliminate this pollutant, such as filtration, sedimentation, and adsorption, which generate fewer toxic by-products. However, these techniques do not

achieve mineralization of the pollutant [1]. In addition, advanced oxidation systems such as electrooxidation, electrocoagulation, chemical coagulation, photodegradation, and coupled systems have also been evaluated (biologically advanced systems) [10–12]. Wastewater treatment requires high energy loads and also chemical compounds, which represent an increase in toxicity of treated effluents [13].

Some strategies with biological treatments, mainly anaerobic systems, have been incorporated into the degradation of reactive anthraquinoic dyes, with elimination reaching 90% with residence times ranging from 8–24 h [14]. The use of these systems for colorant elimination is an economically and environmentally sustainable option. However, only a proportion of pollutants are adsorbed onto the granules' surfaces (the same happens with physical processes), and the absorbed colorant fails to mineralize completely, generating highly toxic degradation by-products [15]. Likewise, biological systems with selective bacteria [2], fungi [16,17], and enzymes [18] have been incorporated to eliminate these pollutants.

In the degradation of reactive anthraquinoic dye using catalysts, enzymes are responsible for hydrolyzing and oxidizing contaminants. The use of specific enzymes for pollutant degradation reduces operating times and costs, and allows optimization of the treatment. Laccases are enzymes commonly called blue multicopper oxidases, and they are the most widely distributed of all the large blue copper-containing proteins [19]. They are produced by a wide variety of microorganisms such as bacteria [20], yeast [21], and fungi [22] isolated from different environments [23]. Basidiomycete and bacterial laccases have been widely reported; however, using these enzymes for dye removal requires the use of redox mediators [19,24] Mtibaa et al. [20] reported ascomycete laccases that do not require the incorporation of this type of mediator. Refs [25–27] Data of laccase production from the genus *Gliomastix* is limited. In Mexico, there are many environments in which to obtain microorganisms capable of producing these kinds of enzyme [21,22].

The challenge in catalyst application in residual effluent treatment is the immobilization and/or trapping of enzymes in a system that facilitates recovery for reuse. Different hydrogels for enzyme immobilization have been used, such as: poly hydroxyethyl methacrylate-co-vinylene carbonate p (HEMMA-co-VC), chitosan [28], genipin-activated chitosan [24], polyacrylamide-alginate cryogel [29], Ca-alginate [25], and others. Galactomannans are polysaccharides used as thickeners and emulsifiers in the food industry, their main advantage is that they are obtained from the endosperm seeds of numerous plants, mainly legumes [26,27], which means that it is an economical option for enzymatic trapping. For this reason, the main objective of this work was to evaluate the degradation of Rb-2 in synthetic water using ascomycete laccase *Gliomastix murorum* sp. HP3 trapped in galactomannan beads.

## 2. Materials and Methods

### 2.1. Materials

Galactomannan polysaccharide (M.W. of approx. 310 kDa) from *Ceratonia siliqua* (algarrobo) seeds, lignin alkali and Reactive Blue 2 (Cibacron Blue 3GA) were purchased from Sigma-Aldrich® (Merck KGaA, Darmstadt, Germany). Agave bagasse, which is an agro-industrial waste derived from agave distillates, was supplied by a local distiller company; before use, it was ground and sieved to a particle size less than 0.149 mm.

### 2.2. Isolation, Screening, and Identification of Laccase-Producing Microorganism

Isolations were made from soil samples from Tamuín, San Luis Potosí, Mexico. Sample suspensions in sterile distilled water with 1% Tween 80 were diluted in a ratio of 10 mL of water for each gram of soil. They were incubated for 1 h at room temperature (25 °C) and agitation (250 rpm). Decimal dilutions were made from $10^{-1}$ to $10{,}000^{-1}$ in sterile 150 mM NaCl, and 100 μL in Petri dishes of 10 cm diameter were inoculated with Küster medium [30], formulated with (g $L^{-1}$): glycerol, 10; casein, 0.3; $KNO_3$, 2; NaCl, 2; $K_2HPO_4$, 2; $MgSO_4 \cdot 7H_2O$, 0.05; $CaCO_3$, 0.02; $FeSO_4 \cdot 7H_2O$, 0.01, and bacteriological agar,

18. Isolated colonies were collected and passed on a fresh agar medium, then biomass was recovered and stored in 20% glycerol at −20 °C.

Screening of laccase-producing strains was performed in agar plates using 1% lignin as the sole carbon source to replace soluble starch in Küster medium. The appearance of a brown halo around the colony was considered positive for laccase production.

The selected strain was identified by molecular analysis. A fresh microorganism culture was used to DNA extraction with a commercially available kit, Dneasy® Plant Mini Kit (Qiagen, Hilden, Germany). The PCR was made using OneTaq® DNA polymerase (New England Biolabs, Ipswich, MA, USA) on a 96-well thermal cycler (Veriti™, Applied Biosystems, Foster City, CA, USA). The primers ITS1 and ITS4 [31] were employed to amplify approximately 600 bp fragments from the transcribed spacer region in the nuclear ribosomal repeat unit. The PCR product was sequenced by Macrogen USA Corp. (Rockville, MD, USA). Consensus sequences were analyzed using the CLC Main Workbench 5.5 software (CLCBio, Aarhus, Denmark). The BLAST algorithm was used to compare obtained sequences with the nucleotide GenBank database [32]. A phylogenetic tree was built where the evolutionary distances were computed using the Maximum Composite Likelihood method [33].

### 2.3. Culture Conditions

The culture medium was optimized for laccase production as described in Supplementary Data. The effect of nitrogen, carbon, and co-substrate was studied by multivariate and surface response experimental designs. The optimized medium contained (g L$^{-1}$): glucose, 12.0; agave bagasse, 28.0; urea, 2.6; NaCl, 2.0; K$_2$HPO$_4$, 2.0; MgSO$_4$·7H$_2$O, 0.05; CaCO$_3$, 0.02; FeSO$_4$·7H$_2$O, 0.01, and was inoculated with $5 \times 10^5$ spores m L$^{-1}$ of a fresh surface culture and incubated at 30 °C in Erlenmeyer flasks with orbital shaking at 250 rpm. Laccase activity was monitored for 120 h. The biomass generation was estimated by weight difference. The reducing sugars quantification was performed using Miller's DNS method [34]. A standard glucose curve was used as a reference.

### 2.4. Enzyme Extraction

The culture of *Gliomastix murorum* was stopped at 72 h to obtain the maximum production of laccase activity. The extracellular extract was recovered by centrifugation at 10,000 rpm for 10 min at 4 °C and frozen at −20 °C until its subsequent use.

### 2.5. Activity Assays

Laccase activity was determined by monitoring the oxidation of 7.5 mM ABTS (2,2′ azino bis (3 ethylbenzthiazoline-6-sulfonic acid)) in 100 mM citrate-phosphate buffer, pH 6 at $\lambda_{max}$ = 405 nm. Assays were carried out in a 96-well plate in 200 μL (estimated 0.6 cm path length) of reaction volume, which was incubated at 37 °C and monitored every 30 s in a microplate spectrophotometer (Epoch™ 2, BioTek™). One unit of enzymatic activity corresponds to 1.0 μmol of ABTS oxidized per minute under test conditions. An ABTS molar extinction coefficient of 36.8 mM$^{-1}$ cm$^{-1}$ was used [35].

Reactive Blue 2 (Rb-2) was evaluated as a substrate for laccase. Enzyme activity was tested under the same pH (6), temperature (37 °C), and stirring conditions as per ABTS. In contrast to this, the Rb-2 assay was monitored at $\lambda_{max}$ = 610 nm. One unit of enzymatic activity corresponds to 1.0 μmol of Rb-2 degraded per minute under test conditions. The molar extinction coefficient came from a standard curve (8.4 mM$^{-1}$ cm$^{-1}$).

### 2.6. Preparation of Galactomannan/Laccase Beads

Galactomannan (1.8 g L$^{-1}$) was dissolved in distilled water. Laccase extract (50:50) $v/v$ was prepared. First, galactomannan solution was heated in a 1500 W microwave oven on high power for 15 s; then when the mix was cold (25 °C), the laccase extract was added. The mixture was stirred using a magnetic stirrer to ensure complete mixing. For bead preparation, the mixture was added dropwise to a beaker containing 100 mL of acetone

at 4 °C, and it was left under constant orbital shaking at 50 rpm for 1 h. The beads were recovered using a strainer and placed on a glass plate for 10 min at 25 °C to ensure complete evaporation of the acetone. The enzymatic activity (EA) was measured in the beads that were obtained. The beads were stored at 4 °C for later use. A schematic illustration of bead preparation is shown in Figure 1.

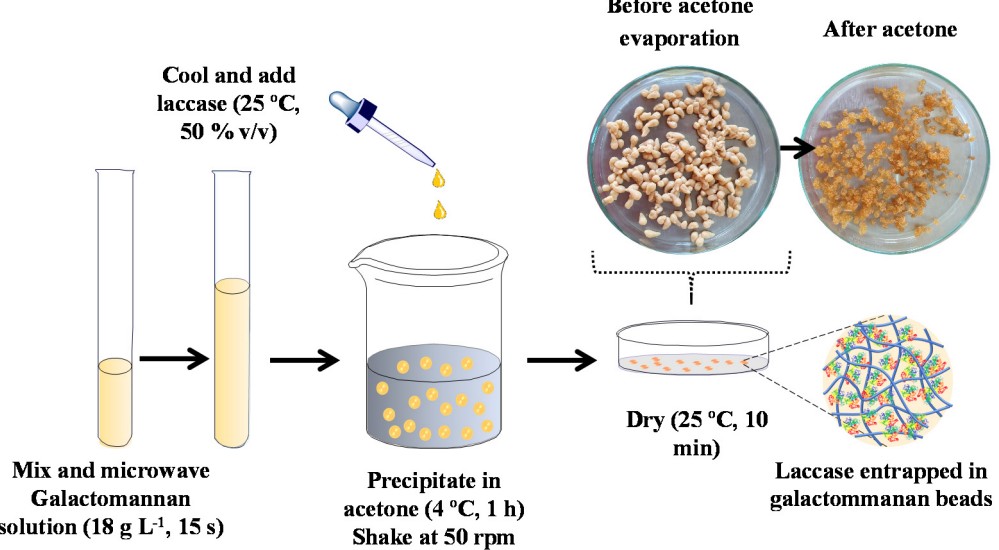

**Figure 1.** Schematic illustration of galactomannan/laccase bead preparation.

### 2.7. Reactive Blue 2 Decolorization Assays

The Rb-2 decolorization by galactomannan entrapped laccase was conducted in conical tubes containing 5.0 mL of 0.5 mM Rb-2 (dissolved in 100 mM citrate-phosphate, pH 6) and 3 beads of entrapped laccase (0.39 mU mg$^{-1}$ of catalyst). A simultaneous experiment with equal activity units of free laccase and the same reaction conditions was performed. Samples were incubated on an orbital shaker at 250 rpm and 37 °C. A sample (200 µL) was taken every 20 min and analyzed spectrophotometrically at $\lambda_{max}$ = 610 nm. The extent of decolorization was expressed in terms of a percentage calculated as follows:

$$Decolorization~(\%) = \frac{(A_0 - A_t)}{A_0} \times 100$$

where $A_0$ is the initial absorbance and $A_t$ is the absorbance of test sample. All samples were analyzed in triplicate.

To determine the possible adsorption of Rb-2 on galactomannan beads, control experiments with galactomannan enzyme free beads where also conducted.

### 2.8. Effect of Temperature and pH on the Enzyme Stability

Thermostability tests of the free and entrapped laccases were carried out by incubating the enzyme at different temperatures, ranging from 4 °C to 80 °C, using 100 mM citrate-phosphate (pH 6). Furthermore, the stability at different pH was analyzed, incubating the enzyme preparations in different 100 mM citrate-phosphate buffer solutions (pH 2–9) at room temperature.

In both studies, the sample was left under assay conditions for 120 min; next, the residual activity was measured using ABTS as the substrate for the reaction at pH 6. Assays were carried out in triplicate on 2 mL microtubes containing 0.5 mL of buffer. Residual activity was calculated considering the maximum activity as 100%. For activity tests, the entrapped laccase was recovered by draining the beads, while the free enzyme was conveniently diluted in 100 mM citrate-phosphate buffer at pH 6.

### 2.9. Reusability Capacity of Entrapped Laccase

The reusability test was evaluated in batch. A bead of entrapped laccase with an initial activity of 0.39 mU mg$^{-1}$ of catalyst and 0.5 mL of 0.5 mM Rb-2 (dissolved in 100 mM citrate-phosphate, pH 6) were placed in 2 mL microtubes. Tubes were incubated on an orbital shaker (250 rpm) at 37 °C for 30 min. Samples were taken at the beginning and after the incubation time to determine the decolorization (%). After one cycle of decolorization, the liquid was withdrawn, and 0.5 mL of fresh Rb-2 solution was added to initiate a new cycle. A total of 7-batch cycles were evaluated. Decolorization was reported as the percentage of dye elimination in every batch test. Dye decolorization in the first cycle was considered as 100%.

## 3. Results and Discussion

### 3.1. Isolation, Screening, and Identification of Laccase-Producing Microorganism

A total of 33 strains were isolated from soil samples from El Tamuín, San Luis Potosí, Mexico. These were screened in a selective medium containing lignin as the sole carbon source. After 14 days, 4 strains had grown, and only the strain HP3 generated a brown halo around the colony, indicating laccase production (Figure S2).

The PCR amplification of the ITS region (476 bp) of HP3 and phylogenetic analysis (Figure 2) revealed that the most closely related fungal strain was *Gliomastix murorum* sp. JCKQF8 (sequence ID: KT968540.1), with 99.58% identity. Ten sequences were selected to make the phylogenetic tree based on, the identity percentage, the score, and the query cover.

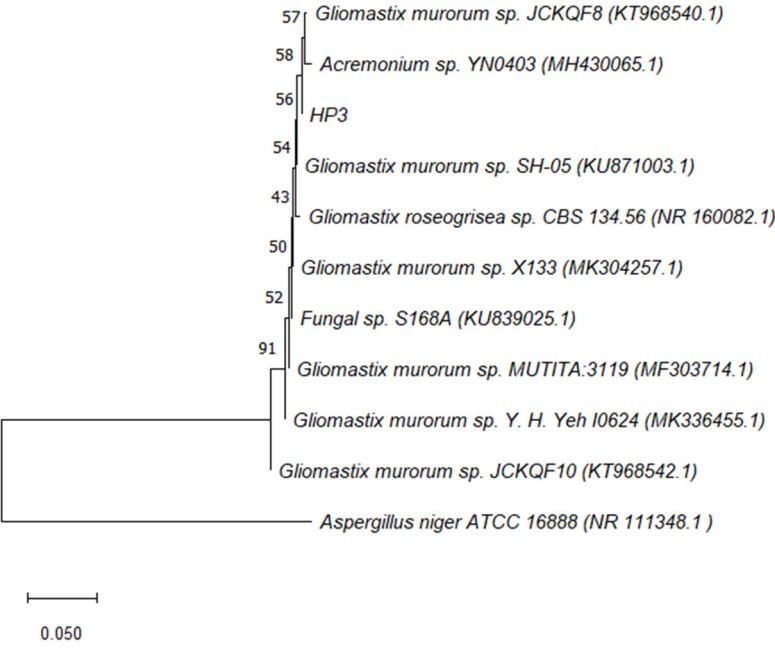

**Figure 2.** Phylogenetic tree based on ITS rRNA gene sequence showing the position of strain HP3 and *Gliomastix murorum* related species. *Aspergillus niger* was used as an out-group. Numbers at the nodes are bootstrap values (of 1000 replicates).

*G. murorum* is a type of saprophyte belonging to the order *Hypocreales*, which are related to the *Dothideales*, *Magnaporthales*, *Capnodiales*, and *Leotiomycetes* as ascomycetous fungi responsible for the degradation of lignocellulosic materials in forest litter and soil [36]. To our knowledge, no laccase activity has been reported in this genus; furthermore, there are few ascomycete laccases reported in the literature, and most of the laccases that have been studied in depth come from the division Basidiomycetes [18].

### 3.2. Laccase Production

*Gliomastix murorum* sp. HP3 was grown in agave bagasse-based medium to produce laccase. The same profile was observed between enzyme production and cell biomass, the maximum enzyme activity was reached at 72 h of culture (1798.6 U L$^{-1}$) (Figure 3). This phenomenon was similar to laccases produced by white-rot fungi grown in natural substrates, such as agricultural residues in submerged fermentation [37,38]. However, Myasoedova et al. [39] observed that the maximum production of laccase activity in *Hypocreal* ascomycetes occurs between three and seven days when they are grown in a mineral medium with a complex carbon source (grain crops or potato).

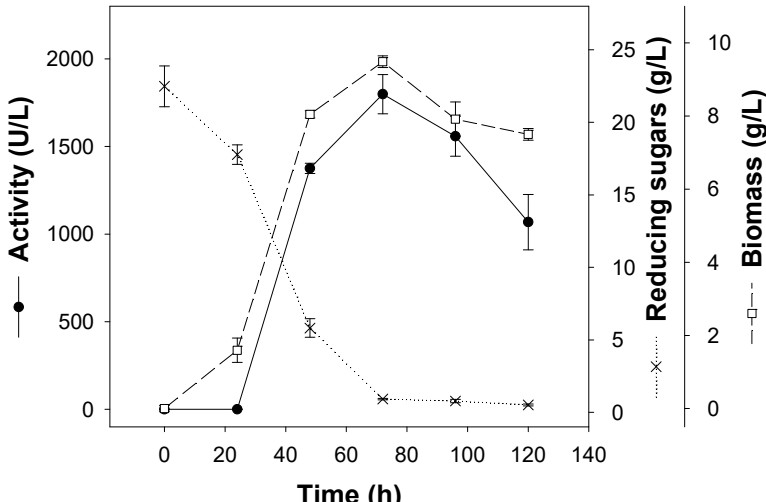

**Figure 3.** Growth kinetics of *Gliomastix murorum* sp. HP3 and its production of extracellular laccases in a culture medium based on agave bagasse.

On the other hand, it can be observed that both the production of laccase and the growth of the strain decline at the time the reducing sugars are exhausted; this can be solved through a fed batch fermentative strategy, which takes into account both the catabolic repression by glucose, as well as the maintenance of the culture [40].

### 3.3. Kinetic Parameters of Free and Entrapped Laccase

The kinetic parameters of the free and immobilized enzyme were determined for the oxidation of 2,2′ azino bis (3 ethylbenzthiazoline-6-sulfonic acid) (ABTS) and Reactive Blue 2 (Rb-2) decolorization. Kinetics were performed using a range of substrate concentrations, and the results obtained are shown in Table 1. Both free and entrapped laccase showed a higher affinity for Rb-2 than for ABTS, while $V_{max}$ was higher in the oxidation of ABTS for entrapped enzyme. The results suggest that the immobilization of laccases in galactomannan beads does not modify the nature of the enzyme; however, the lower catalytic rate can be attributed to diffusion phenomena within the hydrogel, causing low accessibility of the substrate to the active sites [24]. *Gliomastix murorum* sp. HP3-free laccase presented greater affinity for ATBS than entrapped laccase; similar results were found for *Thielavia* sp. ascomycete and *Trametes pubescens* laccase-free [20,24,41]. Nevertheless, $V_{max}$ for ABTS substrate in entrapped laccase was 2.5 times higher than free laccase, and the opposite was true for Rb-2 substrate; $V_{max}$ of 5.67 for free laccase and 1.19 for entrapped laccase, respectively.

**Table 1.** Kinetic parameters of free and entrapped laccases.

| Enzyme | Substrate | Vmax ($\mu$mol min$^{-1}$) * | Km (mM) |
|---|---|---|---|
| Free | | | |
| | ABTS | 3.05 | 0.28 |
| | Rb-2 | 5.67 | 0.18 |
| Entrapped | | | |
| | ABTS | 7.54 | 6.00 |
| | Rb-2 | 1.19 | 1.42 |

* Rate per gram (entrapped) or milliliter (free) of sample.

### 3.4. Evaluation of Laccase in Reactive Blue 2 Decolorization

The effectiveness of free and entrapped laccase in Rb-2 decolorization is shown in Figure 4. Rb-2 was 82% decolorized after 2 h of treatment with entrapped laccase; as expected, the treatment with the free enzyme reached 100% at the same time.

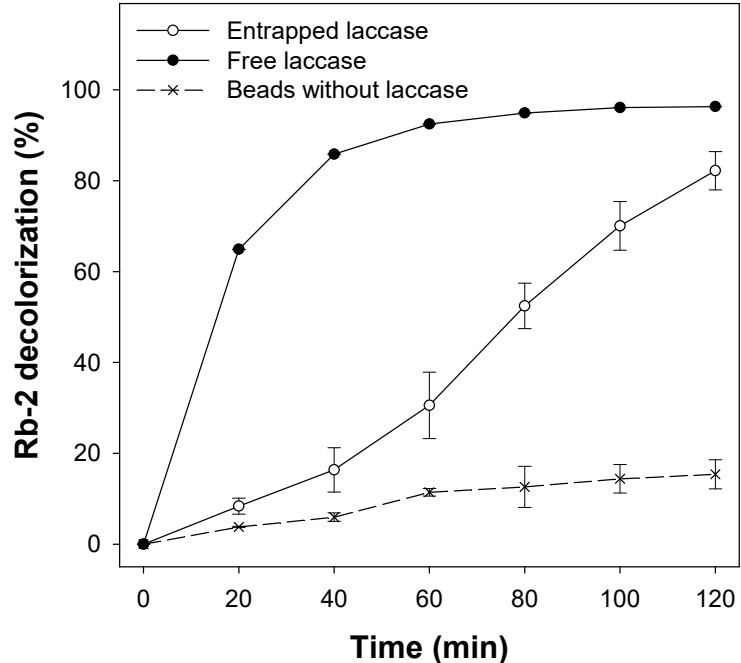

**Figure 4.** Decolorization of Reactive Blue 2 (0.5 mM) by free and entrapped laccase. The kinetics were performed with 0.39 mU mg$^{-1}$ of laccase at 37 °C, pH 6.0, under orbital shaking (250 rpm). The change in absorbance at 610 nm was used to compute the percent decolorization. Galactomannan beads without enzyme was used as control.

The use of galactomannan for enzyme entrapment has not been reported before. Different hydrogels for enzyme immobilization have been used. Bayramoglu et al. [15] immobilized laccase (*Trametes versicolor*) in poly hydroxyethyl methacrylate-co-vinylene carbonate p (HEMMA-co-VC) microbeads with acetosyringone as mediator, finding 43% Cibacron Blue 3GA degradation in a 2 h contact period. Ma et al. [24] found 77.49% removal of the dye Acid Black 172 through laccase (*Trametes pubescens*) immobilized by genipin-activated chitosan beads after 96 h of incubation. Zheng et al. [28] studied Acid Black 172 dye, finding 68.84% decolorization efficiency after a 48 h reaction by laccase (*Trametes pubescens*) immobilized in chitosan beads using glutaraldehyde as crosslinker; both studies used 50 mg L$^{-1}$ as the initial dye concentration. Likewise, polyacrylamide-alginate cryogel was used for laccase (*Trametes versicolor*) immobilization in the elimination of different dyes, and results showed a >75% removal after 5 h (50 mg L$^{-1}$ dye initial concentration) [29].

On the other hand, synthetic mediators are usually applied to improve dye oxidation; however, they are toxic compounds and their application increases dye treatment costs [19]. *Gliomatix murorum* sp. HP3 laccase does not require the addition of mediators to obtain high dye decolorization. In this work, more than 60% decolorization of Rb-2 (0.5 mM initial concentration) was achieved in the first 20 min when free laccase was used, while Othman et al. [42] observed a 29.29% decolorization of Rb-2 (0.12 mM initial concentration) after 35 min reaction using a laccase extract of the Basidiomycete *Agaricus bisporus* CU13, adding HBT 1 mM as mediator.

These results highlight the potential applications that laccase derived from *Gliomatix murorum* sp. HP3 entrapped in galactomannan beads has for dye removal over other hydrogel matrices, with the additional benefit of not requiring the addition of mediators.

### 3.5. Temperature and pH Effects on the Stability of Free and Entrapped Laccase

The effect of temperature (4–80 °C) on free and entrapped laccases was evaluated (Figure 5a). The temperature effect on laccase stability was different from the free and entrapped enzyme. For free laccase, residual activity decreased near 55% after 20 °C and kept decreasing as the temperature increased, while for entrapped laccase, residual activity remained >95% until 40 °C and decreased <50% after 50 °C. These results show an increase in the thermal stability of the enzyme when galactoatches of 30 min were analyzed. Residual activimannan is used. Biopolymer use confers additional resistance and a protective microenvironment matrix.

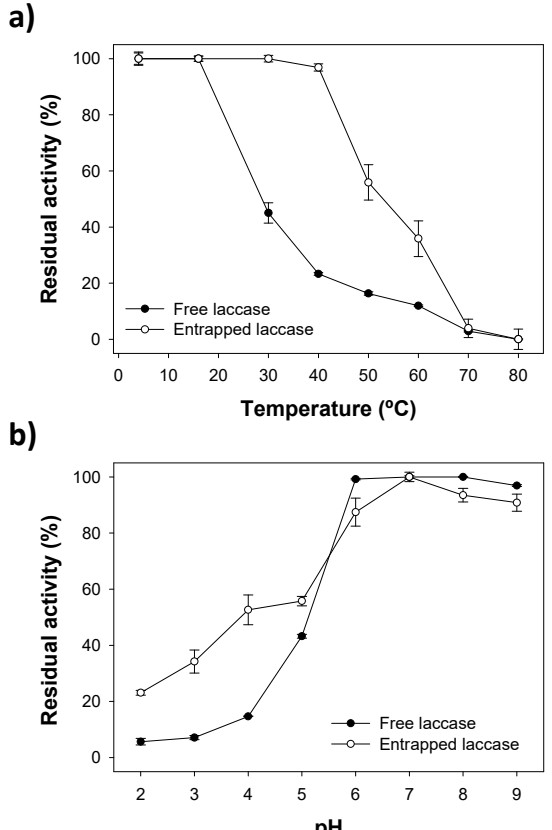

**Figure 5.** Effect of temperature (**a**) and pH (**b**) on stability of free and entrapped laccase. Enzymes were pre-incubated for 2 h under test conditions; after that, their residual activity was measured at pH 6 in Rb-2 as substrate and then calculated based on the activity of non-incubated enzyme as control (100%).

The effect of pH (2–9) on stability of free and entrapped laccases was evaluated (Figure 5b). Entrapped laccase presented higher residual activity (23.75 ± 11.25) to acidic

pH (2–5). Similar residual activity was maintained for free and entrapped laccase to alkaline pH (6–9). Lassouane et al. [41] found high stability under acidic conditions (pH 3–6) in crosslinked-entrapped laccase in Ca-alginate beads, and more than 80% of its initial activity was retained. Results suggest that galactomannan hydrogel protects enzyme against acidic pH.

### 3.6. Reusability of Entrapped Laccase

The capacity of reusability of immobilized laccase is one of the most important indicators in industrial enzymatic applications (for reducing the cost) [41]. Therefore, it is necessary to investigate the feasibility of reusing the biocatalyst immobilized on different substrates [28,43]. Laccases entrapped in galactomannan beads have not been studied before. In this work, seven successive batches of 30 min were analyzed. Residual activity was >90% after four cycles and >80% after six cycles (Figure 6). It is worth noting that, at 30 min of reaction, 20% decolorization was achieved and, to reach an 80% level, 120 min are needed; the difference in operation time could affect the loss of activity, lowering the reusability.

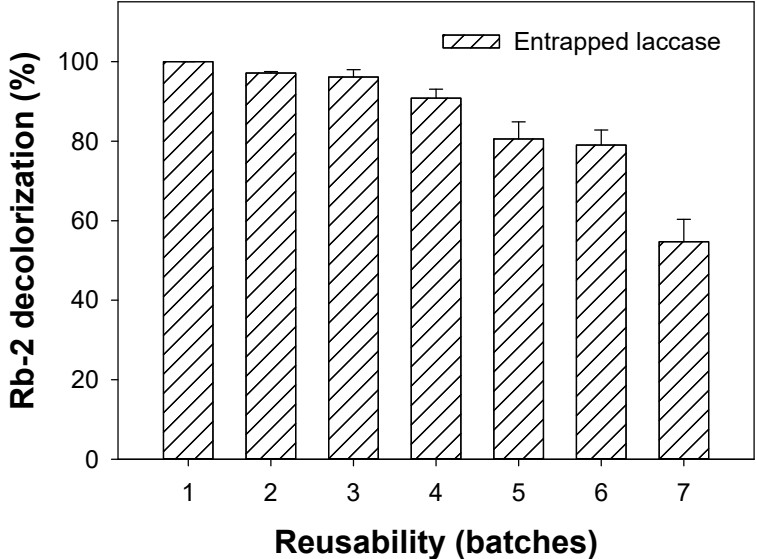

**Figure 6.** Effect of reuse on the activity of entrapped laccase. The assays were performed with 0.39 mU mg$^{-1}$ of laccase at 37 °C, pH 6, under orbital shaking (250 rpm). After one cycle of decolorization, the liquid was withdrawn, and a fresh Rb-2 solution (0.5 mM, pH 6) was added to initiate a new cycle.

Results obtained from the reuse test of laccase entrapped in galactomannans are similar to those reported by Zheng et al. [28], who used chitosan to immobilize the laccase of *Trametes pubescens*, with a residual activity >60% after six cycles. Alginate is another widely used hydrogel for the entrapment of enzymes in the removal of dyes and other compounds. Reda et al. [43] found that the relative activity of the enzyme immobilized in alginate retains approximately 65% of activity after the seventh cycle. Daàssi et al. [25] detected that, after the fourth cycle, the discoloration values were >70% in laccase immobilized in Ca-alginate beads. It is worth mentioning that combinations of different matrices have been made to increase the reuse of immobilized enzymes. Some studies report the combination of graphene oxide in alginate to degrade different compounds, finding decolorizations >75%, with reuse between four and six cycles [36,44]. The decrease in enzyme activity could be attributed to the inactivation and loss of enzyme molecules during each cycle [28]. Furthermore, the environment of the solution could induce some conformational changes in the enzyme during continuous reprocessing, which would cause a decrease in the performance of catalytic activity [45].

The results suggest that the use of galactomannans for laccase entrapment implies excellent operational stability and good reusability.

## 4. Conclusions

The laccase produced by ascomycetes *Gliomastix murorum* sp. HP3 exhibits a greater affinity for Reactive Blue 2 (Rb-2) decolorization, an anthraquinoid dye of significant toxicity, than by 2,2′ azino bis (3 ethylbenzthiazoline-6-sulfonic acid) (ABTS), which is the conventional substrate for laccases. In addition, the catalytic rate is similar to that reported for other laccases that require the addition of redox mediators. The galactomannans matrix generated for the entrapment of laccases conferred enzyme stability against temperature and pH. The use of biopolymer allowed the enzyme to be reused for up six cycles, maintaining 82% in Rb-2 decolorization activity. Using hydrogels based on galactomannans for enzymatic trapping is a viable alternative which offers the advantage of obtaining dye beads that could be stored under ambient conditions.

**Supplementary Materials:** The following are available online at https://www.mdpi.com/article/10.3390/su13169019/s1, Figure S1: Pareto analysis plot of effect of A inductor, B substrate and C nitrogen, on yield of enzyme production, Figure S2: Main effects plot: inductor (agave and cane), substrate (glucose and starch) and nitrogen (potassium nitrate and urea) for enzyme production yield, Figure S3: Interaction plot: Inductor and substrate (upper left), inductor and nitrogen (lower left) and substrate and nitrogen (lower right) on enzyme production yield, Figure S4: Response surface showing the interaction between three parameters and extracellular laccase production, Table S1: Experimental multifactorial design for screening of medium components for *Gliomastix murorum* (strain HP3) laccase production, Table S2: Response surface design.

**Author Contributions:** Conceptualization, R.B.M.-P., J.N.-A., J.A.R., M.M.d.C., M.A.C.-R. and I.C.R.-S.; methodology, R.B.M.-P., R.M.C.-R., A.B.-T., J.E.P.-S., M.O.C.-G. and I.C.R.-S.; software, M.A.C.-R., M.O.C.-G. and I.C.R.-S.; validation, R.B.M.-P., M.A.C.-R. and I.C.R.-S.; formal analysis, M.A.C.-R., R.B.M.-P., M.O.C.-G. and J.A.R.; investigation, M.A.C.-R., M.O.C.-G. and I.C.R.-S.; resources, J.A.R., R.M.C.-R., A.B.-T., M.M.d.C., M.O.C.-G. and M.A.C.-R.; data curation, J.N.-A., J.E.P.-S., R.B.M.-P., M.A.C.-R., A.B.-T and I.C.R.-S.; writing—original draft preparation, R.B.M.-P., M.O.C.-G., M.A.C.-R. and I.C.R.-S.; writing—review and editing, R.B.M.-P., M.O.C.-G., M.A.C.-R. and I.C.R.-S.; visualization, J.N.-A., J.A.R. and R.M.C.-R. All authors have read and agreed to the published version of the manuscript.

**Funding:** This project was financed by the Sectorial Research and Education Fund SEP/CONACyT, Mexico (CB-2016/283183).

**Institutional Review Board Statement:** Not applicable.

**Informed Consent Statement:** Not applicable.

**Data Availability Statement:** Not applicable.

**Acknowledgments:** I.C.R.-S. acknowledges the SEP/CONACyT for her postdoctoral fellowship.

**Conflicts of Interest:** The authors declare no conflict of interest.

## Abbreviations

The following abbreviations are used in this manuscript:

| | |
|---|---|
| ABTS | 2,2′ azino bis (3 ethylbenzthiazoline-6-sulfonic acid) |
| BLAST | Basic Local Alignment Search Tool |
| DNS | dinitrosalisylic acid |
| Rb-2 | Reactive Blue 2 |

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
