# Peer review of "Galactomannans for Entrapment of Gliomastix murorum Laccase and Their Use in Reactive Blue 2 Decolorization"

_sustainability, doi:10.3390/su13169019_

Round 1

Reviewer 1 Report

Romero-Soto, and coworkers reported the discovery of a novel laccase from the ascomycete Gliomastix murorum. Moreover, they tested the possibility of the isolated laccase to be entrapped in galactomannan beads and used for the decolorization of the Reactive blue 2 (Rb-2).

-Albeit the authors tried to highlight many positive aspects of their work, the novelty of this manuscript is real poor, since laccases are largely explored for dye decolorization.

-The author used an agar plate test with lignin as sole carbon source, and the activity against ABTS to prove the isolation of a novel laccase, however they should give other evidences about the identity of the isolated enzyme.

-The manuscript is very badly written rendering difficult its reading.

Minor revision

- The figures shown in the Supplementary materials should have been removed from the manuscript

- An analysis of the pH-activity profile should have been added to justify the use of the ABTS assay at pH6

- Discussion of kinetic parameters (“Both free and entrapped laccase showed a higher affinity for Rb-2 than for ABTS, while Vmax was higher in the oxidation of ABTS for both enzymatic presentations”) is not coherent with data reported in the corresponding Table 1

Author Response

Reviewer 1

Romero-Soto, and coworkers reported the discovery of a novel laccase from the ascomycete Gliomastix murorum. Moreover, they tested the possibility of the isolated laccase to be entrapped in galactomannan beads and used for the decolorization of the Reactive blue 2 (Rb-2).

  • Albeit the authors tried to highlight many positive aspects of their work, the novelty of this manuscript is real poor, since laccases are largely explored for dye decolorization.

Answer: Indeed, laccases are widely explored for the degradation of synthetic dyes, however, in this article it is proposed to obtain an enzyme extract that presents a high catalytic activity without the need to add redox mediating agents. Most of the work that has been reported on laccase degradation has been primarily from basidiomycete enzymes. This work explores the ascomycete laccase fungus, which is easier to grow and can use complex substrates such as agave bagasse, a residue from the tequila industry. On the other hand, to our knowledge, the production of laccases from Gliomastix murorum has not been reported. Likewise, the use of galactomannans for enzymatic trapping has not been reported.

  • The author used an agar plate test with lignin as sole carbon source, and the activity against ABTS to prove the isolation of a novel laccase, however they should give other evidences about the identity of the isolated enzyme.

Answer: The proposal made in this work has aimed at the use of biomass generated as waste in tequila industry, as well as materials from plant species that are little used. likewise, investing in extra step for enzyme purification does not fit to context of proposal, that is why, we decided to use the enzyme extract as is. Therefore, depth biochemical characterization was not done because the laccase was not isolated. In preliminary studies, the activity of other enzymes related to lignin degradation was measured, such as: lignin peroxidase, versatile peroxidase, manganese peroxidase and laccase activity was the majority.

  • The manuscript is very badly written rendering difficult its reading.

Answer: Manuscript was reviewed by a native speaker of the English language.

  • Minor revision

The figures shown in the Supplementary materials should have been removed from the manuscript.

Answer: We appreciate your comment, the suggestions were addressed in the text.

  • An analysis of the pH-activity profile should have been added to justify the use of the ABTS assay at pH6.

Answer: pH 6 was chosen according to a review of literature regarding laccase activity measured in ABTS for other microbial cultures. (ma 2018; mtibaa 2018).

  • Discussion of kinetic parameters (“Both free and entrapped laccase showed a higher affinity for Rb-2 than for ABTS, while Vmax was higher in the oxidation of ABTS for both enzymatic presentations”) is not coherent with data reported in the corresponding Table 1.

Answer:

We appreciate your observation. It is not unusual to find enzymes that have a higher affinity with a substrate despite having a lower catalytic activity (Ma 2018; Lassouane 2019).

Reviewer 2 Report

In this work, the authors studied a novel laccase production from ascomycete Gliomastix murorum in agroindustrial wastes and entrapped in galactomannan beads for Reactive blue 2 (Rb-2) decolorization.  Before the publication, some aspects need minor improvements such as: 
•    Figure S1 S2: why the numeration of the Figures is not continuous? Figures S1 and S2 should be Figure 1 and Figure 3, consequently, the number of other Figures has to change;
•    Introduction: a panoramic on the problem of the wastewater management has to be added, also with examples on several fields (e.g. digestate from anaerobic digestion “Giuliano A, Catizzone E, Freda C, Cornacchia G. Valorization of OFMSW Digestate-Derived Syngas toward Methanol, Hydrogen, or Electricity: Process Simulation and Carbon Footprint Calculation. Processes 2020;8:526. doi:10.3390/pr8050526”;
•    Conclusion: the research application of this study isn’t clear. Authors have to deepen the potential applications of their research;

Author Response

Reviewer 2

In this work, the authors studied a novel laccase production from ascomycete Gliomastix murorum in agroindustrial wastes and entrapped in galactomannan beads for Reactive blue 2 (Rb-2) decolorization.  Before the publication, some aspects need minor improvements such as:

  • Figure S1 S2: why the numeration of the Figures is not continuous? Figures S1 and S2 should be Figure 1 and Figure 3, consequently, the number of other Figures has to change

Answer: We appreciate this observation. It was because supplementary material figures were added to text (as s1, s2, s3 ...), they were already eliminated and order in the document was corrected.

  • Introduction: a panoramic on the problem of the wastewater management has to be added, also with examples on several fields (e.g. digestate from anaerobic digestion “Giuliano A, Catizzone E, Freda C, Cornacchia G. Valorization of OFMSW Digestate-Derived Syngas toward Methanol, Hydrogen, or Electricity: Process Simulation and Carbon Footprint Calculation. Processes 2020;8:526. doi:10.3390/pr8050526”;

Answer: Due to all the reviewers made suggestions about the introduction, it was restructured according to the comments.

  • Conclusion: the research application of this study isn’t clear. Authors have to deepen the potential applications of their research;

Answer: We appreciate your comment, changes have been made in the document.

Reviewer 3 Report

Overall, the manuscript is good. However, several changes in the manuscript are required before recommending acceptance to the journal.

Introduction, method, and discussion sections are very weakly written. Result and conclusion portion is well written. Below are my specific comments regarding the manuscript.

Line 33: These aromatic: The meaning of “these” is not clear

Line 34-35: please correct the typo

Lines 37-41: the meaning of the latter part of the sentence is not clear.

Lines 51-54: Please check the citations. Although this could have been a simple mistake, this has significantly reduced the confidence of the reviewer on the manuscript.

Line 56-58: This statement requires citation

Overall, the introduction section requires a drastic change. Please try to make it succinct and summarize the content required for your study.

Line 108: shaking at 24 g. Please including shaking at RPM.

Line 135: It is understandable that the final Galactomannan concentration will be 0.9% (v/v) but it is not understandable where does Galactomannan (1.8%, v/v) comes from? Line 74 mentions that Galactomannan was purchased from Sigma, does it originally come as 1.8% (v/v)? If not, how was 1.8% Galactomannan made? These things need to be clarified

Same section:

laccase extract (50% v/v) was it the extract mentioned in lines 118-120? Was the extract diluted to obtain 50% v/v solution?

The sample was heated for 15 s. How much volume of the reaction mixture was used? If larger volumes were used, 15s heating might not be enough for the uniform distribution of heat.

I have major issues with Figure 1 and its description in section 2.6. Please include all the steps in the figure and description. This is one of the most important parts of this manuscript. The clarity in this section is a must.

Line 165 and where applicable, please use RPM.

Figure S2: meaning of * is not clear. Does this figure come from the HP3 strain or all 33 strains? Since the manuscript mixes up 33 strains, 4 strains, and HP3, this figure is confusing. Please explain how strain HP3 was identified in much clear way.

Line 216 and where applicable: please italicize the scientific name.

Figure 2: Please present the alignment. As there is 99.58% identity among HP3 and Gliomastix murorum, it is also important to explain how those 9 sequences were selected

Figure 4: it is evident that the same level of activity as free laccase was not obtained by using the entrapped laccase. With this, the impact of this work is reduced. The authors need to offer some valid statements for entrapment in the manuscript. A simple explanation can be reusability and temperature stability.

Figure numbers in the Main text file and Supplementary file are different.

Author Response

Reviewer 3:

  • Overall, the manuscript is good. However, several changes in the manuscript are required before recommending acceptance to the journal. Introduction, method, and discussion sections are very weakly written. Result and conclusion portion is well written. Below are my specific comments regarding the manuscript.

Answer: We appreciate your comments and suggestions.

  • Line 33: These aromatic: The meaning of “these” is not clear

Answer: Due to all the reviewers made suggestions about the introduction, it was restructured according to the comments.

  • Line 34-35: please correct the typo

Answer: We appreciate your comment, changes have been made in the document.

  • Lines 37-41: the meaning of the latter part of the sentence is not clear.

Answer: The introduction was improved, so that paragraph was restructured.

  • Lines 51-54: Please check the citations. Although this could have been a simple mistake, this has significantly reduced the confidence of the reviewer on the manuscript.

Answer: We appreciate your observation, changes have been corrected in the document.

  • Line 56-58: This statement requires citation.

Answer: Introdution was changed and some citation were added.

  • Overall, the introduction section requires a drastic change. Please try to make it succinct and summarize the content required for your study.

Answer: We appreciate your observation, introduction was reestructured.

  • Line 108: shaking at 24 g. Please including shaking at RPM.

Answer: Changes have been made in line 92, 117, 144, 156, 180.

  • Line 135: It is understandable that the final Galactomannan concentration will be 0.9% (v/v) but it is not understandable where does Galactomannan (1.8%, v/v) comes from? Line 74 mentions that Galactomannan was purchased from Sigma, does it originally come as 1.8% (v/v)? If not, how was 1.8% Galactomannan made? These things need to be clarified.

Answer:

Galactomannan is in powder and 18 g/L of concentration was used, and it was dissolved in distilled water: laccase extract (50:50) v / v. Corrections have been made in the text.

  • Same section: Laccase extract (50% v/v) was it the extract mentioned in lines 118-120? Was the extract diluted to obtain 50% v/v solution?

Answer: Distilled water: laccase extract (50:50) v/v was used to dissolve the galactomannans. Corrections have been made in the text.

  • The sample was heated for 15 s. How much volume of the reaction mixture was used? If larger volumes were used, 15s heating might not be enough for the uniform distribution of heat.

Answer:For beads production, only 50% of the mixture was heated (galactomannan and distilled water; 25 mL), once cold (25 ° C), extract was added (25 mL), in order not to damage the enzyme activity.

  • I have major issues with Figure 1 and its description in section 2.6. Please include all the steps in the figure and description. This is one of the most important parts of this manuscript. The clarity in this section is a must.

Answer: Suggestions were attended.

  • Line 165 and where applicable, please use RPM.

Answer: Changes have been made in the text.

  • Figure S2: meaning of * is not clear. Does this figure come from the HP3 strain or all 33 strains? Since the manuscript mixes up 33 strains, 4 strains, and HP3, this figure is confusing. Please explain how strain HP3 was identified in much clear way.

Answer: The procedure for strain HP3 is detailed in lines 103-106. Missing information was added to the foot of figure S2 and a paragraph was added at the beginning of the supplementary material.

  • Line 216 and where applicable: please italicize the scientific name.

Answer: We appreciate your observation, changes have been corrected in the document.

  • Figure 2: Please present the alignment. As there is 99.58% identity among HP3 and Gliomastix murorum, it is also important to explain how those 9 sequences were selected

Answer: 10 sequences were selected to make the phylogenetic tree considering the identity percentage, the score, and the query cover. The information was added to lineas 195-197. Next we present the multiple sequence aligment.

https://drive.google.com/file/d/1AFWAwJbCXUOXfjtSdfPh7H8UjO_LGHnH/view?usp=sharing

  • Figure 4: it is evident that the same level of activity as free laccase was not obtained by using the entrapped laccase. With this, the impact of this work is reduced. The authors need to offer some valid statements for entrapment in the manuscript. A simple explanation can be reusability and temperature stability.

Answer: The immobilization technique by entrapment, usually reduces the catalytic activity, however, it gives it properties of resistance to high temperatures, acidic pH and it can be reused in different cycles. This is discussed in lines 285-287 and in the conclusion.

  • Figure numbers in the Main text file and Supplementary file are different.

Answer: We appreciate your observation, corrections have already been made on the document and supplementary material.

Reviewer 4 Report

The need to decolorize textile dyes is presented. In Introduction, you explain the toxicity of these dyes. If the chemical structure of blue2 will be modified with the treatment of laccase enzyme and the toxicity of day is also assumed to be reduced? Is there some literature about this since you have no experiments about toxicity.

In introduction use, the reference system claimed by this journal also in line 53! See the guidance!

Line 55: Rhodococcus!

Materials and methods: How did you isolate lignin and how did you analyze it in order to know that you had 1 % lignin (line 89)? Did you assume that a specific amount of agave bagasse is lignin? What about cane line 211? Or could you omit both Figures S1 and S2, since Fig. S1 is between materials and methods and in both Figs. Present the most essential details as text or as a Table. For  S2 you do not describe the experiment used glucose or starch as a carbon source and lignin as the other carbon source, which you call as inducer. Is it causing induction so that the laccase enzymes will be forming?  

In Fig S2 “the effect” is too unspecific as the title. If you keep these supplement figures in the manuscript you must describe the experiment in materials and methods! You have those in supplementary material but also in this material, there is no description of the true materials and methods. The true materials and methods should be so clear that another scientist could repeat the experiments but now I should guess very much since many important details are lacking.

Describe also better and in more detailed the media which you used to study laccase at different pH values and with different carbon sources!  

Explain all abbreviations when they are in text the first time. Now at least ABTS is impossible to guess.

Results and discussion:  The long paragraph starting in line 275 is more a list of different studies and their results and less a discussion given views to your work. Think what you wish to say here! Divide the paragraph into shorter ones!   

Explain better the benefits of entrapping the laccase! When testing the stability of the enzyme could you have some control? what about reusability? What was the sample used in reusability tests?

In references, there are no research works with the name et al. Add all authors! Year with bold!

Author Response

Reviewer 4

  • The need to decolorize textile dyes is presented. In Introduction, you explain the toxicity of these dyes. If the chemical structure of blue2 will be modified with the treatment of laccase enzyme and the toxicity of day is also assumed to be reduced? Is there some literature about this since you have no experiments about toxicity.

Answer: Bayramoglu et al (2019), studied Cibacron Blue 3GA biodegradation by immobilized laccase and conducted toxicity evaluation by Daphnia magna and Chlorella vulgaris, finding that, when there is a short treatment time (from 0 to 60 min) effluents treated are more toxic. After 60 min enzymatic treatment period, the dye completely depredated and any toxic compound was not detected by the MALDI-ToF-MS studies.

  • In introduction use, the reference system claimed by this journal also in line 53! See the guidance!

Answer:We appreciate your observation, changes have been corrected in the document.

  • Line 55: Rhodococcus!

Answer: Suggestion was attended.

  • Materials and methods: How did you isolate lignin and how did you analyze it in order to know that you had 1 % lignin (line 89)? Did you assume that a specific amount of agave bagasse is lignin? What about cane line 211? Or could you omit both Figures S1 and S2, since Fig. S1 is between materials and methods and in both Figs. Present the most essential details as text or as a Table. For  S2 you do not describe the experiment used glucose or starch as a carbon source and lignin as the other carbon source, which you call as inducer. Is it causing induction so that the laccase enzymes will be forming?  

Answer: Figures S1, S2, and S3 were passed to complementary material, in which the experiment carried out for optimization of culture medium is described. Regarding lignin, it was used commercially, which is already isolated and comes from an alkali extraction process. In line 86 of section 2.1 of materials, the origin of the lignin was added.

  • In Fig S2 “the effect” is too unspecific as the title. If you keep these supplement figures in the manuscript you must describe the experiment in materials and methods! You have those in supplementary material but also in this material, there is no description of the true materials and methods. The true materials and methods should be so clear that another scientist could repeat the experiments but now I should guess very much since many important details are lacking.

Answer: In complementary material the requested information has already been added, likewise figures S1, S2, and S3 were passed to complementary material.

  • Describe also better and in more detailed the media which you used to study laccase at different pH values and with different carbon sources!  

Answer: We appreciate your observation. Conditions of factor used in design were added on Table 1 in complementary material

  • Explain all abbreviations when they are in text the first time. Now at least ABTS is impossible to guess.

Answer: Suggestions were attended. Changes have been made in line 128-129.

  • Results and discussion:  The long paragraph starting in line 275 is more a list of different studies and their results and less a discussion given views to your work. Think what you wish to say here! Divide the paragraph into shorter ones!   

Answer: Ideas of that paragraph were organized giving greater meaning to the text.

  • Explain better the benefits of entrapping the laccase! When testing the stability of the enzyme could you have some control? what about reusability? What was the sample used in reusability tests?

Answer: The benefits of laccase entrapment are described in lines 285-287 and in the conclusion. Likewise, to determine the possible adsorption of Rb-2 on galactomannan beads, control experiments were performed with enzyme-free galactomannan beads (lines 164-165).

  • In references, there are no research works with the name et al. Add all authors! Year with bold!

Answer: Format of references was homogenized according to journal style.

Round 2

Reviewer 1 Report

Although the authors have slightly modified the manuscript to address some issues, there are still unresolved questions.

  • Even if laccase activity measurements for other microbial cultures at pH 6 towards ABTS are reported in literature, only a pH-activity profile can be useful in defining the optimal pH for the assay (notably for claimed novel laccase)
  • I am sure that it is not unusual to find enzymes that have a higher affinity with a substrate despite having a lower catalytic activity. The point is that the sentence “[…] while Vmax was higher in the oxidation of ABTS for both enzymatic presentations.” is in contradiction with the data reported in the Table 1, where Vmax was lower in the oxidation of ABTS for the free enzyme.
  • This previously described behaviour also affects the rate of the decolourization of the dye of the free enzyme with respect to the entrapped one, thus rendering almost unnecessary the entrapment. Neither the reusability is helpful, since the reported experiment include cycle of 30 mins (seven), though after 30 mins the decolorization is about 20% since the entrapped laccase needs 120 mins to achieve about 80% of decolourization.

Author Response

-Although the authors have slightly modified the manuscript to address some issues, there are still unresolved questions.

We are grateful with the reviewer for their comments we resolved the question they asked for.

-Even if laccase activity measurements for other microbial cultures at pH 6 towards ABTS are reported in literature, only a pH-activity profile can be useful in defining the optimal pH for the assay (notably for claimed novel laccase)

We agree with the reviewer comment, enzyme activity was not measured at the enzyme optimal pH, we measured the effect of pH on stability but not on activity, we rewrite the paragraph to clarify that. We were not performing a complete biochemical characterization of the enzyme (this work is actually on curse, to be published soon). In the actual work our goal was to highlight the use of galactomannans as laccases entrappers and their effect on stability.

Line 293 “The effect of pH (2.0 – 9.0) on stability of free and entrapped”

-I am sure that it is not unusual to find enzymes that have a higher affinity with a substrate despite having a lower catalytic activity. The point is that the sentence “[…] while Vmax was higher in the oxidation of ABTS for both enzymatic presentations.” is in contradiction with the data reported in the Table 1, where Vmax was lower in the oxidation of ABTS for the free enzyme.

We are very grateful with the reviewer, we rewrite the sentence to correct the mistake as follows “while Vmax was higher in the oxidation of ABTS for entrapped enzyme”

-This previously described behaviour also affects the rate of the decolourization of the dye of the free enzyme with respect to the entrapped one, thus rendering almost unnecessary the entrapment. Neither the reusability is helpful, since the reported experiment include cycle of 30 mins (seven), though after 30 mins the decolorization is about 20% since the entrapped laccase needs 120 mins to achieve about 80% of decolourization.

We agree with the reviewer respect to decolorization is faster with the free enzyme, but the entrapped one can be reused meanwhile the free enzyme cannot. In the other hand, the loss of activity in the reusing is principally affected by washing steps, that is why the protocol implemented include 30 mins cycles in order to reach more cycles to be studied despite not achieving 80% decolorization. We added a paragraph in results to clarify the point that reviewer remark.

Line 314 “It is worth noting that at 30 min of reaction, 20 % of decolorization was achieved, to reach an 80 % 120 min are needed, the operation time could affect the loss of activity lowering the reusability.”

Reviewer 3 Report

The authors did an excellent job revising the manuscript

Author Response

The authors did an excellent job revising the manuscript

We are grateful to the reviewer for his comments, each observation helped to improve the writing.

Reviewer 4 Report

The authors have made the most corrections.

Please, in lines 61 and 262 fulfill the sentences:

Line 61: oxidases, and they are …

Line 262: however, they …

Line 70 the first bacterium is Rhodococcus (not Rhadocccus). It is related to Mycobacterium.

See from the guidance of the journal Sustainability how the references must be made. There is no et. al. Use italics for the name of journal and volume! Use bold for the year.

The reference list should include the full title, as recommended by the ACS style guide. Style files for Endnote and Zotero are available.

References should be described as follows, depending on the type of work:

  • Journal Articles:
    1. Author 1, A.B.; Author 2, C.D. Title of the article. Abbreviated Journal NameYearVolume, page range.
  • Books and Book Chapters:
    2. Author 1, A.; Author 2, B. Book Title, 3rd ed.; Publisher: Publisher Location, Country, Year; pp. 154–196.
    3. Author 1, A.; Author 2, B. Title of the chapter. In Book Title, 2nd ed.; Editor 1, A., Editor 2, B., Eds.; Publisher: Publisher Location, Country, Year; Volume 3, pp. 154–196.
  • Unpublished materials intended for publication:
    4. Author 1, A.B.; Author 2, C. Title of Unpublished Work (optional). Correspondence Affiliation, City, State, Country. year, status (manuscript in preparation; to be submitted).
    5. Author 1, A.B.; Author 2, C. Title of Unpublished Work. Abbreviated Journal Name year, phrase indicating stage of publication (submitted; accepted; in press).
  • Unpublished materials not intended for publication:
    6. Author 1, A.B. (Affiliation, City, State, Country); Author 2, C. (Affiliation, City, State, Country). Phase describing the material, year. (phase: Personal communication; Private communication; Unpublished work; etc.)
  • Conference Proceedings:
    7. Author 1, A.B.; Author 2, C.D.; Author 3, E.F. Title of Presentation. In Title of the Collected Work(if available), Proceedings of the Name of the Conference, Location of Conference, Country, Date of Conference; Editor 1, Editor 2, Eds. (if available); Publisher: City, Country, Year (if available); Abstract Number (optional), Pagination (optional).
  • Thesis:
    8. Author 1, A.B. Title of Thesis. Level of Thesis, Degree-Granting University, Location of University, Date of Completion.

The paper Sustainability 20157(2), 2189-2212; https://doi.org/10.3390/su7022189  has got the most citations of this journal. Check their references!

Author Response

-The authors have made the most corrections.

Please, in lines 61 and 262 fulfill the sentences:

Line 61: oxidases, and they are …

Line 262: however, they …

We appreciate your observation, the corrections have been made in the text.

-Line 70 the first bacterium is Rhodococcus (not Rhadocccus). It is related to Mycobacterium.

Revisor are absolutely right, we appreciate your observation, the changes were made.

-See from the guidance of the journal Sustainability how the references must be made. There is no et. al. Use italics for the name of journal and volume! Use bold for the year.

The reference list should include the full title, as recommended by the ACS style guide. Style files for Endnote and Zotero are available.

Changes have been made to the document